# Residual Lung Function Impairment Is Associated with Hyperventilation in Patients Recovered from Hospitalised COVID-19: A Cross-Sectional Study

**DOI:** 10.3390/jcm10051036

**Published:** 2021-03-03

**Authors:** Ernesto Crisafulli, Daniele Gabbiani, Giulia Magnani, Gianluigi Dorelli, Fabiana Busti, Giulia Sartori, Gianenrico Senna, Domenico Girelli, on behalf of the RESPICOVID Study Investigators

**Affiliations:** 1Respiratory Medicine Unit, Department of Medicine, University of Verona and Azienda Ospedaliera Universitaria Integrata of Verona, 37126 Verona, Italy; giulia.sartori.verona@gmail.com; 2Department of Medicine, Section of Internal Medicine, University of Verona and Azienda Ospedaliera Universitaria Integrata of Verona, 37126 Verona, Italy; daniele.gabbiani@yahoo.com (D.G.); giuliamagnani1@gmail.com (G.M.); fabiana.busti@univr.it (F.B.); domenico.girelli@univr.it (D.G.); 3School of Medicine in Sports and Exercise, University of Verona, 37129 Verona, Italy; gianluigi.dorelli@gmail.com; 4Department of Medicine, Allergy and Clinical Immunology Section, University of Verona and Azienda Ospedaliera Universitaria Integrata of Verona, 37126 Verona, Italy; gianenrico.senna@univr.it

**Keywords:** COVID-19, lung function, hypocapnia, hyperventilation, restrictive pattern, interstitial pneumonia

## Abstract

Patients who have recovered from COVID-19 show persistent symptoms and lung function alterations with a restrictive ventilatory pattern. Few data are available evaluating an extended period of COVID-19 clinical progression. The RESPICOVID study has been designed to evaluate patients’ pulmonary damage previously hospitalised for interstitial pneumonia due to COVID-19. We focused on the arterial blood gas (ABG) analysis variables due to the initial observation that some patients had hypocapnia (arterial partial carbon dioxide pressure-PaCO_2_ ≤ 35 mmHg). Therefore, we aimed to characterise patients with hypocapnia compared to patients with normocapnia (PaCO_2_ > 35 mmHg). Data concerning demographic and anthropometric variables, clinical symptoms, hospitalisation, lung function and gas-analysis were collected. Our study comprised 81 patients, of whom 19 (24%) had hypocapnia as compared to the remaining (*n* = 62, 76%), and defined by lower levels of PaCO_2_, serum bicarbonate (HCO^3−^), carbon monoxide diffusion capacity (DL_CO_), and carbon monoxide transfer coefficient (K_CO_) with an increased level of pH and arterial partial oxygen pressure (PaO_2_). K_CO_ was directly correlated with PaCO_2_ and inversely with pH. In our preliminary report, hypocapnia is associated with a residual lung function impairment in diffusing capacity. We focus on ABG analysis’s informativeness in the follow-up of post-COVID patients.

## 1. Introduction

A total of eighty-seven percent of patients who have recovered from COVID-19 show persistent symptoms at 60 days, particularly fatigue and dyspnea [1]. Lung function impairments with a reduction of the carbon monoxide diffusion capacity (DL_CO_) has been evident at discharge [2] and during the early convalescence phase of patients recovered from COVID-19 [3]; the carbon monoxide transfer coefficient (K_CO_) alterations might better characterise the gas exchange efficiency [4]. Few data are available evaluating a more extended period of COVID-19 clinical progression, particularly the oxygenation level and the ventilatory pattern at rest. Our preliminary report focused on variables related to the arterial blood gas (ABG) analysis and characterised patients with specific alterations. This interest raised from the initial observation that some post-COVID patients had hypocapnia, documented by lower levels (≤35 mmHg) of arterial partial carbon dioxide pressure (PaCO_2_). Any technical bias has been excluded, and several patients repeated blood sample to confirm the results. Aim of our study was, therefore, to characterise patients with hypocapnia (PaCO_2_ ≤ 35 mmHg) in comparison to patients with normocapnia (PaCO_2_ > 35 mmHg). 

## 2. Materials and Methods

At our tertiary hospital (AOUI Verona, Verona, Italy), we have organised a dedicated outpatient clinic for adult patients previously hospitalised for interstitial pneumonia due to COVID-19, with or without respiratory failure. The RESPICOVID study, a prospective observational trial, has been designed to comprehensively evaluate the prevalence, clinical impact, and predictive factors of pulmonary damage in patients recovered from COVID-19. Due to the explorative nature of our study and the feasibility of observations derived, here we report some aspects related to preliminary data. For this reason, no formal sample size calculation was made and we had not a comparison group of not exposed (controls without COVID-19). The study has been described according to a cross-sectional design, following the STROBE guidelines. The local Ethics Committee approved the study protocol (No. 2785CESC), according to the Good Clinical Practice recommendations and the requirements of the Declaration of Helsinki. Written informed consent was obtained from all participants.

All consecutive patients discharged were considered, excluding those unable to reach the outpatient service (e.g., permanently bedridden) or perform tests (e.g., severe cognitive impairment). All measures were collected prospectively beginning on 6 July 2020, 4 months after patients’ discharge (median time 127 days, standard deviation [SD] 18 days). The ABG analysis has been performed in all patients enrolled, at rest, in-room air and quiet condition. 

We recorded demographic and anthropometric variables, clinical symptoms, laboratory, and gas-analysis. Values of arterial partial oxygen pressure (PaO_2_) were also standardised (*st*PaO_2_) according to the lower levels of PaCO_2_ (<40 mmHg) by the following formula: *st*PaO_2_ = [(PaCO_2_ · 1.66) + PaO_2_] − 66.4.

Data from hospitalisation for COVID-19 were also compared [5].

Lung function was performed according to the international recommendations [6]; a flow-sensing spirometer connected to a computer for data analysis (Jaeger MasterScreen PFT System) was used for the measurements. Forced vital capacity (FVC), forced expiratory volume in the first second (FEV_1_), and total lung capacity (TLC) were recorded. FEV_1_/FVC ratio was taken as the index of airflow obstruction. DLCO and KCO were measured by the single breath method. FEV_1_, FVC, TLC, DLCO and KCO were expressed as percentages of the predicted values [7,8]. 

Data are reported with numbers (percentages) for categorical variables, mean (SD) or median (first quartile; third quartile) for continuous variables with a normal or non-normal distribution, respectively. A preliminary Shapiro–Wilk test was performed. Categorical variables were compared by the χ^2^ test or the Fisher exact test, while continuous variables were assessed by the independent *t*-test or the non-parametric Mann–Whitney test. Pearson (r) and Spearman (ρ) correlations have been carried-out between variables. All analyses were performed using IBM SPSS, version 25.0 (IBM Corp., Armonk, NY, USA), with *p* values of <0.05 considered statistically significant. 

## 3. Results

Our preliminary report comprised 81 patients, of whom 19 (24%) had hypocapnia as compared to the remaining (*n* = 62, 76%), and defined by lower levels of PaCO_2_, serum bicarbonate (HCO^3−^), DL_CO_, and K_CO_ with an increased level of pH and arterial partial oxygen pressure (PaO_2_). Comparing data from hospitalisation, hypocapnic patients had lower PaCO_2_ and HCO^3−^ (Table 1). K_CO_ was directly correlated with PaCO_2_ and inversely with pH; PaCO_2_ was directly correlated with PaCO_2_ at hospitalisation (Figure 1).

## 4. Discussion

Although lung function alterations in term of diffusion impairment in our patients are in line with other reports [2,3], our is the first report examining ABG in patients with COVID-19 pneumonia in a long interval from hospital discharge (>4 months). We report that nearly one-fourth of patients recovered from an interstitial pneumonia COVID-19 have hypocapnia, present at the moment of hospitalisation and associated with a restrictive residual pattern.

Several pulmonary disorders such as pneumonia, pulmonary embolism, asthma or interstitial lung disease (ILD) may produce hypocapnia; the leading physiologic cause of the reduction of PaCO_2_ is related to hyperventilation [9]. Severe interstitial pneumonia due to COVID-19 cause hypoxaemia with mechanisms related to intrapulmonary shunt (oedema and atelectasis), loss of lung perfusion regulation, intravascular microthrombi and impaired diffusion capacity [10]. Although shunt and a low ratio of the alveolar ventilation to blood flow (V_A_/Q) mismatch are the most frequent causes of hypoxaemia, diffusion limitation may cause hypoxemia also in the absence of V_A_/Q mismatch [11]. Moreover, lung units with hypoperfusion due to intravascular deficits increase the alveolar dead space with poor ventilation [11]: as a consequence, to maintain a normal arterial PaCO_2_ the respiratory drive increase the minute ventilation primarily by increasing tidal volume (hyperpnea) and respiratory rate (tachypnea) [10,11]. At hospitalisation for COVID-19, 37 of our patients (58%) had a severe hypoxaemic respiratory failure (PaO_2_/FiO_2_ ≤ 300) but with no residual hypocapnia differences. Similarly, the correction of oxygen pressure by the *st*PaO_2_, such as other data concerning oxygenation at hospitalisation and in the recovery phase seems to be little associated with hypocapnia. Mechanical adaptation to the increased elastic work of breathing or a reflex-mediated originating from the chest wall receptors may be a resting hyperventilation mechanism, which probably starts in our patients from hospitalisation. However, we cannot exclude the possibility of an increased virus-induced respiratory drive stimulation, explaining the long-time persistence of hypocapnia. Notably, hyperventilation has no impact on clinical symptoms perceived from patients and concerning in particular extensional dyspnea; other causes of symptom perception will probably have to be sought. 

In other ILD conditions, not COVID-19, the association of hypocapnia regardless hypoxemia has been reported. Hypocapnia induced by resting hyperventilation has been observed in asbestos-exposed subjects with a restrictive disorder related to mild pleural plaques [12]. Moreover, in patients having ILD associated with polymyositis and dermatomyositis, hypocapnia’s presence increases the probability of a worse prognosis in 6 months [13]. A restrictive ventilatory defect with a reduction of DLCO has been documented in patients recovered from COVID-19 [2,3], closely related to the severity of infection [3]. In our patients, the presence of hypocapnia is associated with a residual lung function impairment in term of diffusing capacity. 

Our major strengths are the originality of considerations, the prospective and consecutive data collection. As limitations, we should mention the relatively small sample size and controls missing: our considerations need to be confirmed in a large study cohort with matched-controls without COVID-19. 

## 5. Conclusions

In conclusion, our report highlights the hypocapnia as a marker of residual lung function impairment of patients recovered from hospitalised COVID-19; moreover, we focus on ABG analysis’s informativeness in these patients’ follow-up. 

## Figures and Tables

**Figure 1 jcm-10-01036-f001:**
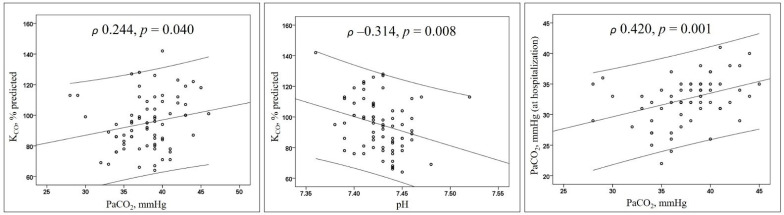
Scatterplots. Lines represent the regression with the 95% confidence intervals. Abbreviations: K_CO_ represents the carbon monoxide transfer coefficient; PaCO_2_, arterial partial carbon dioxide pressure.

**Table 1 jcm-10-01036-t001:** Characteristics of patients considered.

Variables	All Patients (*N* = 81)	Patients with Normocapnia (*N* = 62)	Patients with Hypocapnia (*N* = 19)	*p* Value
Age, years	66.5 ± 11.2	67.2 ± 10.7	64.4 ± 12.5	0.353 ^a^
Male, *n* (%)	54 (67)	39 (63)	15 (79)	0.194 ^b^
BMI, kg∙m^2^	27 (24.4–30)	27.1 (24.3–30.7)	26.9 (25–29.2)	0.705 ^c^
Smoking habit, no/current or former, *n* (%)	46 (57)/35 (43)	37 (60)/25 (40)	9 (47)/10 (53)	0.343 ^b^
Arterial hypertension, *n* (%)	46 (57)	34 (55)	12 (63)	0.522 ^b^
Diabetes mellitus, *n* (%)	10 (12)	8 (13)	2 (10)	>0.999 ^d^
Cough, *n* (%)	9 (11)	8 (13)	1 (5)	0.678 ^d^
Extensional dyspnea, *n* (%)	16 (20)	13 (21)	3 (16)	0.751 ^d^
Asthenia, *n* (%)	27 (33)	20 (32)	7 (37)	0.711 ^b^
Muscle fatigue, *n* (%)	19 (24)	15 (24)	4 (21)	>0.999 ^d^
Heart rate, bpm	69.8 ± 10.5	69.9 ± 10.3	69.8 ± 11.4	0.967 ^a^
Respiratory rate, bpm	16 (14–20)	16 (14–20)	16 (13.5–20)	0.543 ^c^
SpO_2_, %	97 (96,97)	97 (96,97)	97 (96–98)	0.093 ^c^
pH	7.43 (7.41–7.44)	7.42 (7.41–7.44)	7.44 (7.43–7.46)	**0.002** ^**c**^
PaCO_2_, mmHg	38 (36–40)	39 (37–41)	34 (32–34)	**<0.001** ^**c**^
PaO_2_, mmHg	96.3 ± 12.8	94.1 ± 12.3	103.2 ± 12.1	**0.007** ^**a**^
PaO_2_/FiO_2_	458.3 ± 61	448.3 ± 58.8	491.2 ± 57.9	**0.007** ^**a**^
PaO_2_/FiO_2_ ≤ 400 ^e^, *n* (%)	15 (18)	14 (23)	1 (5)	0.173 ^d^
*st*PaO_2_, mmHg	91.7 ± 11.9	91.9 ± 12.2	91.3 ± 11.2	0.884 ^a^
P(A-a)O_2,_ mmHg	6.04 ± 12.1	6.29 ± 12.5	5.25 ± 11.2	0.747 ^a^
HCO^3−^, mmol/L	25 ± 2	25.8 ± 1.6	22.7 ± 1.4	**<0.001** ^**a**^
C-reactive protein, mg/L	1 (1,2)	1 (1,2)	1 (1–4)	0.926 ^c^
Haemoglobin, g/dL	14.1 ± 1.5	14.1 ± 1.5	14 ± 1.7	0.907 ^a^
Platelets, 10^9^/L	209.8 ± 48.7	211.2 ± 52.3	204.9 ± 33.7	0.664 ^a^
Leucocytes, 10^9^/L	6.1 ± 1.5	5.98 ± 1.5	6.13 ± 1.3	0.710 ^a^
FEV_1_, % predicted	120 ± 21.1	122.2 ± 21.6	112.4 ± 17.6	0.093 ^a^
FVC, % predicted	120 (110.2–140.7)	121 (112–141)	118 (96–139)	0.226 ^c^
FEV_1_/FVC, %	101 (98–106)	102 (98–107)	100 (94–103)	0.073 ^c^
TLC, % predicted	100.5 ± 13.3	101 ± 13.7	98.7 ± 12.1	0.547 ^a^
DL_CO_, % predicted	86.4 ± 16.5	88.6 ± 16.3	78.5 ± 15.3	**0.034** ^**a**^
K_CO_, % predicted	94.9 ± 17.2	97.2 ± 17.4	86.5 ± 13.6	**0.032** ^**a**^
Variables measured at hospitalisation for COVID-19				
pH ^f^ (*N* = 64)	7.47 (7.44–7.50)	7.48 (7.45–7.50)	7.47 (7.43–7.50)	0.394 ^c^
PaCO_2_, mmHg ^f^ (*N* = 64)	33 (31–35)	34 (32–35)	31 (27–33)	**0.010** ^**c**^
PaO_2_/FiO_2_ ^f^ (*N* = 64)	288.8 ± 86.6	284.9 ± 90.5	301.5 ± 73.5	0.522 ^a^
PaO_2_/FiO_2_ ≤ 300 ^f^ (*N* = 64), *n* (%)	37 (58)	31 (63)	6 (40)	0.110 ^b^
*st*PaO_2_/FiO_2_^f^ (*N* = 64)	234.4 ± 86.4	236.3 ± 92.8	228.2 ± 63.7	0.752 ^a^
P(A-a)O_2,_ mmHg ^f^ (*N* = 64)	46.1 (37.4–66.5)	45.7 (36.2–66.5)	46.5 (37.5–69.7)	0.890 ^c^
HCO^3−^, mmol/L ^f^ (*N* = 64)	23.9 ± 2.9	24.7 ± 2.1	21.6 ± 3.8	**<0.001** ^**a**^
Length of stay, days	8.1 (5–13.5)	8 (5–14)	11 (5–13)	0.369 ^c^
Unit of admission, ICU/medical ward, *n* (%)	12 (15)/69 (85)	10 (16)/52 (84)	2 (10)/17 (90)	0.722 ^d^
Pulmonary embolism, *n* (%)	3 (4.5)	3 (5.7)	0 (0)	>0.999 ^d^
Oxygen-therapy, *n* (%)	52 (64)	40 (64)	12 (63)	0.914 ^b^
Lopinavir/ritonavir, *n* (%)	60 (74)	44 (71)	16 (84)	0.372 ^d^
Hydroxychloroquine, *n* (%)	65 (80)	48 (77)	17 (89)	0.335 ^d^
Antibiotics, *n* (%)	22 (27)	16 (26)	6 (32)	0.621 ^b^
Tocilizumab, *n* (%)	16 (20)	11 (18)	5 (26)	0.511 ^d^
Steroids, *n* (%)	22 (27)	14 (23)	8 (42)	0.094 ^b^
Prophylactic LMWH, *n* (%)	26 (41)	22 (45)	4 (27)	0.208 ^d^

Data are shown as mean ± standard deviation, median [25°–75° percentiles], or number (percentages). In bold, significant variables. ^a^ calculated by the independent *t*-test; ^b^ calculated by the χ^2^ test; ^c^ calculated by the non-parametric Mann–Whitney test; ^d^ calculated by the Fisher exact test; ^e^ Defined as the normality limit; ^f^ Data collected at admission in the emergency room and in spontaneous breathing. Atrial fibrillation, ischemic heart disease, vascular disease, chronic heart failure, cancer, chronic kidney disease, liver disease, chronic obstructive pulmonary disease, asthma, and connective-disease have a prevalence of <10% with no significant difference between groups. Abbreviations: BMI define body mass index; SpO_2_, oxygen saturation by pulse oximetry; PaCO_2_, arterial partial carbon dioxide pressure; PaO_2_, arterial partial oxygen pressure; *st*PaO_2_, standardised arterial partial oxygen pressure; PaO_2_/FiO_2_, the ratio of the partial pressure of arterial oxygen to the fraction of inspired oxygen; P(A-a)O_2_, alveolar-arterial gradient; HCO^3−^ serum bicarbonate; FEV_1_, forced expiratory volume at 1^st^ second; FVC, forced vital capacity; TLC, total lung capacity; DL_CO_, diffusion capacity for carbon monoxide; K_CO_, carbon monoxide transfer coefficient; ICU, intensive care unit; LMWH, low molecular weight heparin.

## Data Availability

The data presented in this study are available on request from the corresponding author.

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
