# Peer review of "Residual Lung Function Impairment Is Associated with Hyperventilation in Patients Recovered from Hospitalised COVID-19: A Cross-Sectional Study"

_jcm, 2021, doi:10.3390/jcm10051036_

Round 1

Reviewer 1 Report

The present article report interesting findings on the outpatients COVID-19 patients. The article needs to be imporved with more data on ABG such as Lactate level, P/F, Arteriolar/alveolar gradient, and more data on the clinical course of the disease 

Author Response

We thank the Reviewer for his/her suggestions. We have added data concerning the P/F and the alveolar-arterial gradient, but unfortunately, we have no lactate level data. Concerning the clinical course, we have only data about pulmonary embolism and ICU admission, indirect variables of complicated COVID.  

Reviewer 2 Report

This submission does not pose any particular problem. However, I suggest this document to be written according to the STROBE guidelines in order to respect the methodological standards related to this type of study.

Author Response

We thank the Reviewer for his/her suggestion. We have added in the text some aspects concerning the STROBE guidelines.

Reviewer 3 Report

Thank you for asking me to review this article entitled “Residual lung function impairment is associated with hyperventilation in patients recovered from hospitalized COVID-19: a preliminary report”.

This article reports preliminary data from a study on the lung function of patients with COVID-19 pneumonia in a long interval from hospital discharge. The argument is very important, the aim of the study is formulated, the paper is well written. 

I have just minor observations:

- clinical symptoms are not commented on in the discussion. 

- sentence at line 130 is not clear to me

I think that you can consider this paper for publication in the Journal of Clinical Medicine

Author Response

We thank the Reviewer for his/her positive comments and fascinating suggestions. We have added a sentence reporting the no impact of hyperventilation on clinical symptoms. In line 130, we report the causes of hypoxemia in patients with interstitial pneumonia and the relationships with the possible mechanisms of hyperventilation. We have modified the text, and we hope that now may be more precise. 

Reviewer 4 Report

This is an interesting study about association of residual lung function impariment with hyperventilation in COVID patients.

My major concern relies on statistical analysis with implications for conclusions of the study. A Bonfeeroni correction should be applied for all analyses.

Other informations are needed to draw firm conclusions:

  • Ct thorax. is there a correlate with lung lesion ? or is this impairment in diffusion without radiographic correlate
  • I advise 6 Min walk test to assess functional impairment
  • The data about treatment during the COVID infect should be mentioned (Therapy)

Author Response

We thank the Reviewer for his/her suggestions. We have compared patients with normocapnia and hypocapnia according to the sample distribution (t-test or Mann-Whitney); Bonferroni correction is one of several methods used to counteract the problem of multiple comparisons, and in our study sample we have only two groups. We agree with the suggestion about CT thorax and 6MWT; unfortunately, in our patients, we are unable to have these variables for the analysis. On the contrary, we have added some data concerning the treatment during the COVID hospitalization.      

Round 2

Reviewer 2 Report

Thank you. There are no more major problems except :

Pease insert in MM that you followed Strobe guidelines 

Sample size information is missing: Explain how the study size was calculated and arrived at this number. Why write line 171 "As limitations, we should mention the single-centre and the relatively small sample size. “

Has the protocol been submitted to clinicaltrials.gouv?

You write line 93 A preliminary Shapiro-Wilk test was performed. Correct, except that the test was performed after the description of the variables. Put this sentence after "or non-normal distribution, respectively". 

Explain why this is a preliminary report. Will there be a protocol difference between the two or is there a difference in sample size?

Table 1: Can we consider that all quantitative variables presented as +/- sd means have a normal distribution? Put in footnote the tests performed for each variable by adding exponentially " a, b, c, ... " to the number of the p-value column.

Line 176. You write "Further studies need to understand the pathophysiology of its alterations better. “. Unless I'm mistaken, won't it be the presentation of the final study that will bring added value?

Author Response

Dear Reviewer,

We are thanked for considering our manuscript again for the review process.

Please find below the point-by-point reply to each comment and remark.

  1. Q) Pease insert in MM that you followed Strobe guidelines.
  2. A) We thank the Reviewer for his/her suggestion. We have added this aspect in Material and Methods.

  1. Q) Sample size information is missing: Explain how the study size was calculated and arrived at this number. Why write line 171 "As limitations, we should mention the single-centre and the relatively small sample size. “
  2. A) We thank the Reviewer for his/her suggestion. In the revised version, we have added the sentence: “Due to the exploratory nature of our study…no formal sample size calculation was made”. Moreover, we have added a sentence about our limitation, removing the single-centre.

  1. Q) Has the protocol been submitted to clinicaltrials.gouv?.
  2. A) We thank the Reviewer for his/her suggestion. No, we have not submitted it to clinicaltrial.gov.

  1. Q) You write line 93 A preliminary Shapiro-Wilk test was performed. Correct, except that the test was performed after the description of the variables. Put this sentence after "or non-normal distribution, respectively"..
  2. A) We thank the Reviewer for his/her precious suggestion. We have modified accordingly.

  1. Q) Explain why this is a preliminary report. Will there be a protocol difference between the two or is there a difference in sample size?
  2. A) We thank the Reviewer for his/her suggestion. In the revised version of the manuscript, we have added a sentence about the preliminary report's explanation. We hope that it may be in accord with the Reviewer.

  1. Q) Table 1: Can we consider that all quantitative variables presented as +/- sd means have a normal distribution? Put in footnote the tests performed for each variable by adding exponentially " a, b, c, ... " to the number of the p-value column.
  2. A) We thank the Reviewer for his/her suggestion. We have modified accordingly.

  1. Q) Line 176. You write "Further studies need to understand the pathophysiology of its alterations better. “. Unless I'm mistaken, won't it be the presentation of the final study that will bring added value?
  2. A) We thank and we agree with the Reviewer for his/her suggestion. We have removed the sentence reported.